# Phenolic Profile, Toxicity, Enzyme Inhibition, In Silico Studies, and Antioxidant Properties of *Cakile maritima* Scop. (Brassicaceae) from Southern Portugal

**DOI:** 10.3390/plants9020142

**Published:** 2020-01-22

**Authors:** Chloé Placines, Viana Castañeda-Loaiza, Maria João Rodrigues, Catarina G. Pereira, Azzurra Stefanucci, Adriano Mollica, Gokhan Zengin, Eulogio J. Llorent-Martínez, Paula C. Castilho, Luísa Custódio

**Affiliations:** 1Centre of Marine Sciences, Faculty of Sciences and Technology, University of Algarve, Ed. 7, Campus of Gambelas, 8005-139 Faro, Portugal; c.placines@gmail.com (C.P.); vianacastanedaloaiza@gmail.com (V.C.-L.); mary_p@sapo.pt (M.J.R.); cagpereira@ualg.pt (C.G.P.); 2Department of Pharmacy, University “G. d’Annunzio” of Chieti-Pescara, 66100 Chieti, Italy; a.stefanucci@unich.it (A.S.); a.mollica@uncih.it (A.M.); 3Department of Biology, Science Faculty, Selcuk University, Campus, 42250 Konya, Turkey; gokhanzengin@selcuk.edu.tr; 4Department of Physical and Analytical Chemistry, Faculty of Experimental Sciences, University of Jaén, Campus Las Lagunillas, E-23071 Jaén, Spain; ellorent@ujaen.es; 5CQM—Centro de Química da Madeira, Universidade da Madeira, Campus da Penteada, 9020-105 Funchal, Portugal

**Keywords:** bioactive plant-derived products, tyrosinase inhibitors, salt tolerant plants

## Abstract

*Cakile maritima* Scop. (sea rocket) is an edible halophyte plant with several ethnomedicinal uses. This work reports the chemical profile and bioactivities of food grade extracts from sea rocket organs. Toxicity was determined on mammalian cells, and phenolic profiling and the quantitation of the main metabolites were made by high-performance liquid chromatography coupled to mass spectrometry (HPLC-MS). Enzymatic inhibition was determined towards acetyl- and butyrylcholinesterase (AChE, BuChE), α-glucosidase, α-amylase, and tyrosinase. Docking studies were performed to tyrosinase, on the major metabolites, and samples were tested for antioxidant properties. Extracts were not toxic, were constituted mainly by flavonoids, and some compounds (roseoside and oleuropein) are here described for the first time in the species. The aerial organs’ ethanol extract had relevant activity towards 2,2-diphenyl-1-picrylhydrazyl [DPPH, half maximal inhibitory concentration (IC_50_) = 0.59 mg/mL], and ferric-reducing activity power (FRAP, IC_50_ = 0.99 mg/mL). All samples were more active towards AChE than on BuChE. The ethanol fruits’ extract inhibited α-glucosidase [2.19 mmol of equivalent of acarbose (ACAE)/g]. Samples were active against tyrosinase, especially the aerial organs’ ethanol extracts [25.9 mg of equivalent of kojic acid (KAE)/g]. Quercetin and kaempferol glycosides fit well into the enzymatic pocket of tyrosinase. Our results suggest sea rocket as a candidate to be further explored as a source of bioactive products.

## 1. Introduction

Research on bioactive natural products has focused mainly in marine organisms, such as micro- and macro-algae and marine invertebrates, and in terrestrial glycophyte plants. Halophytes, which are naturally salt-tolerant plants, represent about 2% of terrestrial plant species and are present in about half the higher plant families. They exhibit a high diversity of plant forms and are currently considered an important reservoir of bioactive molecules with multiple biotechnological applications, ranging from food to cosmetic ingredients [1,2,3]. There are reports of the traditional medicinal uses of 43 families, comprising more than 180 halophytic species, in the Mediterranean, the Arabian Sea, and Syria regions [1,4,5]. These uses are wide and include, for example, the treatment of pain, fever, liver and digestive disorders, skin, respiratory, genito-urinary conditions, microbial and parasitic infections, inflammation, dermatitis, wounds, and burns [1,4,5].

*Cakile maritima* Scop. (sea rocket) is an edible halophytic succulent annual herb of the *Brassicaceae* (mustard) family. It is widespread throughout the world in sandy coastal regions, from northern Norway to the southern coasts of Australia [6,7]. Sea rocket grows as a small shrub up to 40 cm high and has a multi-branched stem. The fleshy leaves are alternate with long pinnate lobes and the inflorescences are white, lilac-coloured, or purple, and develop in summer months. Fruits (pods) are small (1.5–2 cm), flat, segmented, and contain 1–2 seeds [6,8]. Sea rocket is traditionally used as a flavouring agent, and young raw leaves may be added to salads, while dried ground roots can be mixed with cereal flours to make bread [9]. This species also has a number of ethnomedicinal uses, for example, as antiscorbutic, digestive, diuretic, and anti-dandruff [8,10]. Sea rocket produces several bioactive metabolites, such as phenolic acids, ascorbic acid, hydrocarbons, sterols, and flavonoids (e.g., quercetin, quercetin-3-*O*-rhamnoside, kaempferol-7-*O*-glucoside, luteolin, luteolin-7-*O*-glucoside, and caffeic acid), and terpenoids, mainly sesquiterpenoids [6,10,11,12,13,14,15,16].

There are some reports of the biological activities of sea rocket extracts. For example, hydroethanolic extracts from stems, seeds, and leaves from plants collected in Italy displayed in vitro radical scavenging activity towards 2,2-diphenyl-1-picrylhydrazyl (DPPH), antimicrobial, anti-inflammatory, and antiproliferative properties [10]. In another work, the salinity effect on the phenolic content and antioxidant activity of methanol extracts of sea rocket leaves, from Tunisia, was reported [12]. Moreover, antibacterial and antioxidant activities were detected in hydromethanolic extracts of leaves from sea rocket collected in Brittany (France) [14]. The present work aimed to expand existing knowledge on the potential use of sea rocket as a source of bioactive products. For that purpose, food grade extracts (ethanol, acetone and water) were made from dried biomass of fruits and aerial vegetative organs (leaves and stems) from plants collected in Southern Portugal, and evaluated for possible toxic effects on three mammalian cell lines. Samples were then chemically profiled by high-performance liquid chromatography coupled to mass spectrometry (HPLC-MS), and evaluated for in vitro enzymatic inhibitory properties towards key enzymes in human diseases, namely Alzheimer’s disease (AD), (acetyl- and butyrylcholinesterase, AChE, BuChE), Type-2 diabetes (α-glucosidase and α-amylase), and hyperpigmentation (tyrosinase). Docking studies were then performed towards tyrosinase on the main metabolites identified in the extracts, and finally, the in vitro antioxidant capacity was evaluated by methods targeting free radicals and metal ions.

## 2. Results and Discussion

### 2.1. Possible Toxic Effects of the Extracts

This work used ethanol, acetone, and water to obtain natural extracts from dried biomass of fruits and aerial vegetative organs (leaves and stems) from sea rocket plants collected in Southern Portugal. These are food grade solvents, and can be used during the processing of raw materials, foodstuffs, food components, or food ingredients (Directive 2009/32/EC of the European Parliament and of the Council of 23 April).

Despite the theoretical concept that natural products are safer than synthetic counterparts, it is well known that plants may contain toxic molecules. Therefore, the evaluation of possible toxic effects of a natural product is required to assure its safety and potential interest to be further explored for biotechnological uses. Different methods are used for this purpose, including the in vitro assessment of acute toxic effects on mammallian cell lines. Following this, the sea rocket extracts were appraised for cytotoxicity towards murine RAW 264.7 macrophages, human embryonic HEK 293 kidney, and human hepatocellular HepG2 carcinoma cell lines. The fruit extracts were generally less toxic than those from the aerial organs, which may be related with the overall lower level of total individual phenolics content, and only the acetone extracts resulted in cellular viabilities lower than 80% (RAW cells: 73.1%, HEK: 79.4%, Table 1). For the aerial organs, viabilities were lower than 80% in RAW macrophages for ethanol (67.7%) and water (72.3%) extracts, in HEK 293 cells for acetone (72.2%), water (76.5%), and ethanol (78.4%) extracts, and in HepG2 cells for acetone (78.3%) (Table 1). In a previous report, no significant toxicity was observed on non-tumoral lymphocytes after treatment with 70% ethanol extracts from different organs of sea rocket, while a strong decrease in cellular viability was obtained in a human multiple myeloma cell line (U266 cells, [10]). A reduced or nil cytotoxicity on in vitro models using mammalian cell lines is indicative of low toxicity of a natural extract [17,18]. Therefore, acccording to our results, the tested extracts have reduced toxicity and may be further explored as innovative natural products with relevant biological properties.

### 2.2. Chemical Composition of the Extracts: Qualitative Profiling

The phytochemical qualitative profile of sea rocket’s extracts is depicted in Table 2. Compounds were numbered by their order of elution keeping the numbering constant in all extracts, and were identified using analytical standards and bibliographic information (Table 2). The chromatogram obtained for the water extract of sea rocket’s fruits is shown in Figure 1, as an example. The chemical composition of the extracts depended on the type of extract and organ (Table 2).

Twenty-five compounds were characterized in the analyzed extracts, most of them quercetin glycosides (compounds **10**, **11**, **12**, **15**, **18**, and **21**) and kaempferol glycosides (compounds **13**, **14**, **16**, **24**, and **25**) (Table 2, Figure 1 and Figure 2). Both types of compounds were generally present in both types of organs, and for all of them, the aglycones were observed at *m*/*z* 301 for quercetin (fragment ions at *m*/*z* 271, 179, and 151) and *m*/*z* 285 for kaempferol (fragment ions at *m*/*z* 255 and 151). The neutral losses of 308, 162, and 146 Da were indicative of rutinoside, hexoside, and deoxyhexoside moieties, respectively. Both quercetin and kaempferol (Figure 2) were identified by comparison with analytical standards. Similarly, compound **26**, with [M-H]^−^ at *m*/*z* 461, was characterized as isorhamnetin-*O*-deoxyhexoside due to the neutral loss of 146 Da and the presence of isorhamnetin at *m*/*z* 315 (fragment ion at *m*/*z* 300).

Quercetin glycosides and kaempferol glycosides are flavonoids, a group of compounds associated with several health-promoting effects, like preventing cardiovascular complications and cancer, and are important ingredients in several products, including nutraceuticals, cosmetics, and medicinal commodoties [19]. Quercetin derivatives, such as glycosides, are present in different plant-derived food foodtuffs like tea, wine, onions, apples, and broccoli, and can prevent and treat non-infectious chronic diseases, such as diabetes, obesity, and hyperlipidemia [20,21]. Kaempferol glycosides also display interesting biological properties, including cytotoxicity toward human cancer cells, free radical scavenging activity, and inhibition of pancreatic lipase [22]. Quercetin and its derivatives, including quercetin-3-O-rhamnoside, were previously identified in aerial parts of sea rocket [3,19]. Likewise, kaempferol and kaempferol glycosides were already detected in the same species, including kaempferol 7-O-rhamnoside, kaempférol 4′-O-glucoside, and kaempférol3-O-glucoside de 7-O-rhamnoside [23].

Compound **1** suffered the neutral loss of 36 Da (HCl) to yield a disaccharide (two hexoside moieties) at *m*/*z* 341. Its characterization was based in the neutral loss of 162 Da and the fragment ions characteristic of hexoside moieties. Compound **7** was also tentatively characterized as a hexoside derivative.

Two glucosinolates—isomers of dihydrogluconapin (compounds **2** and **5**)—were characterized based on bibliographic information [24], and were present in both organs. Glucosinolates are naturally occurring S-linked glucosides characteristic in *Brassicaceae* species [25]. They originate different metabolites by enzymatic hydrolises, including isothiocyanates, which are likely linked to antimicrobial properties [25]. To our best knowledge, these compounds have not been previously reported in sea rocket.

Compound **8** was characterized as roseoside (Figure 2; formate adduct), also known as vomifoliolglucoside or drovomifoliol-*O*-*β*-*D*-glucopyranoside, and was only detected in the extract from aerial organs. Roseoside is a megastigmane glycoside, is present in different plant species, such as *Ludwigia stolonifera* [26], and is first described here in sea rocket. This compound has several biological properties, including in vitro and in vivo anticarcinogenic and in vitro hypertensive [27].

Compound **20** was identified as oleuropein by comparison with an analytical standard. It was only detected in fruits and is firstly described here in the species. Oleuropein is a secoiridoid, one of the major compounds present in the olive fruit, and is endowed with several functional properties, such as anticancer, antibacterial, antifungal, antiparasitic, and antiplatelet aggregation [28].

Compound **22** suffered the neutral loss of 224 Da (591→367), which corresponds to sinapic acid. In addition, the MS^3^ [591→223] fragmentation correlates with sinapic acid. We tentatively characterized this compound as disinapoyl-hexoside, which was only detected in the aerial organs. Similarly, compound **23** was characterized as trisinapoylgentiobioside, and was only present in fruits. Disinapoyl-hexoside and trisinapoylgentiobioside are sinapic acid derivatives, characteristic in the Brassicaceae family [29]. The most important characteristic property of sinapic acid derivatives is their antioxidant capacity, and thus, have potential use in in the food processing, cosmetics, and the pharmaceutical industry [30].

Compounds **27** and **28** were present in both organs and were characterized as the oxylipins oxo-dihydroxy-octadecenoic acid and trihydroxy-octadecenoic acid, respectively. Oxylipins are secondary metabolites resulting from the oxidative metabolism of polyunsaturated fatty acids (PUFAs) through the addition of oxygen. The production of these compounds increases in response to herbivory, pathogenesis, and wounding, and therefore, several oxylipins add for the regulation of innate defense-related and developmental processes in plants, having, for example, antimicrobial or insecticidal properties [31].

### 2.3. Chemical Composition: Quantitative Profiling of Phenolic Compounds

The phytochemical quantitative profile of sea rocket’s extracts is presented in Table 3. Twelve phenolic compounds were quantified in the extracts of sea rocket: ten flavonoids, oleuropein, and disinapoyl-hexoside. Total individual phenolic content (TIPC) was defined as the sum of all the individual compounds that were quantified by HPLC with diode array detection (HPLC-DAD): flavonoids at 350 nm, oleuropein at 240 nm and disinapoyl at 320 nm. The aerial organs’ extracts presented higher TIPC (acetone: 3.94 mg/g dried extract (DE), water: 7.92 mg/g DE and ethanol: 11.5 mg/g DE) than the fruits’ extracts (acetone: 1.28 mg/g DE, ethanol: 4.42 mg/g DE and water: 5.48 mg/g DE). The total phenolic content of leaves of sea rocket was assayed elsewhere by a spectrophotometric assay [30], showing values of 5 to 7 mg gallic acid equivalent per g DE, which are similar to the ones obtained here by liquid chromatography. In all extracts, the majority of the content corresponded to flavonoids (mainly quercetin and kaempferol glycosides), which represented more than 90% of TIPC. This predominance of flavonoids is in agreement with a previous report on *C. maritima* [16]. Quercetin glycosides were present in similar amounts in both organs while kaempferol glycosides were generally higher in the aerial organs. For both compounds, the highest concentrations were found in ethanol and water, whereas acetone extracts presented the lowest concentrations in all cases.

### 2.4. Enzyme Inhibitory Activities

Molecules from natural sources have a high resemblance and binding potential with biological structures, increasing the odds of an interaction with different biological targets [32]. Those targets includes for example enzymes related with human diseases, such as AChE and BuChE for Alzheimer’s disease (AD), α-glucosidase and α-amylase for Type 2 Diabetes Mellitus (T2DM), and tyrosinase, for hyperpigmentation disorders. Several drugs with enzymatic inhibitory properties towards such enzymes are currently in clinical use to treat the mentioned disorders (e.g., acarbose for T2DM, galantamine for AD, and kojic acid for hyperpigmentation). Nevertheless, their potential undesirable side effects (e.g., gastrointestinal disturbances and hepatotoxicity) promotes a growing interest by the pharmaceutical and cosmetic industries to identify innovative, more efficient, and less toxic natural enzymatic inhibitors. In this work, we evaluated the enzyme inhibitory effects of the sea rocket extracts towards the above-mentioned enzymes, and the results are depicted in Table 4.

Samples had similar capacities to inhibit AChE (aprox. 1.3 mg GALAE/g), except the water extracts from aerial vegetative organs, with reduced activity (0.53 mg GALAE/g), but were less active towards BuChE (from 0.26 to 0.92 mg GALAE/g). AD is one of the major causes of dementia. It is a multifactorial disease whose progression is linked to a considerable loss of the neurotransmitter acetylcholine (ACh), which is responsible for memory and cognition, to the accumulation of β-amyloid protein and to the formation of neurotoxic neurofibrillary tangles. The inhibition of AChE not only adds to the maintenance of normal ACh levels, but also decreases β-amyloid accumulation and neurotoxic fibrils formation, being considered as the main therapeutic approach to manage AD symptoms. To the best of our knowledge, this is the first report of the cholinesterase inhibitory properties of sea rocket.

The extracts had low inhibitory activity towards α-amylase (0.02 to 0.26 mmol ACAE/g extract). The same was observed for α-glucosidase, except for the ethanol extract from fruits that displayed a significant inhibition (2.19 mmol ACAE/g). A similar result towards the latter enzyme was observed for hydroethanolic stems extracts from the same species [10]. Our results suggests that the the highest α-glucosidase inhibitory activity displayed by the ethanol extract from fruits may be related with other compounds, rather than those identified in this work. While α-amylase hydrolyzes polysaccharides to oligosaccharides, α-glucosidase hydrolizes oligosaccharides to monosaccharides. Therefore, inhibition of these enzymes decreases the digestion of carbohydrates and the postprandial rise of blood glucose levels after a mixed carbohydrate meal, and is an important therapeutical instruments for the control of T2DM [33,34,35,36]. Moreover, hypoglycemic products with slight amylase inhibitory activity but significant glucosidase inhibitory properties are favored, as an excessive inhibition of amylase often results in gastrointestinal disorders [37]. Therefore, our results suggests ethanol extracts of sea rocket’s fruits as candidates to further studies exploiting its possible antidiabetic assets.

Aditionally, all tested samples, except for water extract from fruits, showed a potent inhibition of tyrosinase. The best results were observed with extracts from the aerial vegetative organs, with values ranging from 19.9 mg KAE/g (water extracts), to 25.9 mg KAE/g (ethanol). Tyrosinase is a copper containing enzyme involved in melanogenesis and therefore, it is a key target in the search for depigmenting agents that can be used in the treatment of epidermal hyperpigmentation disorders, such as lentigo, melasma, age spots, inflammatory hypermelanosis, and trauma-induced hyperpigmentation [38,39,40]. Moreover, tyrosinase catalyzes the oxidation of phenolic compounds to quinones, which results in the enzymatic browning of plant-derived foods [41]. Quinones have an unpleasant flavour and color and react with proteins, reducing its digestibility and the bioavailability of essential amino acids, with consequent nutritional and economical losses during food storage [41,42]. Tyrosinase is also involved in melanogenesis and cuticle formation in insects [43]. Hence, the use of tyrosinase inhibitors as food preservatives, especially from natural sources, is therefore considered as an alternative approach not only to prevent the enzymatic browning of plant-derived foods but also in insect pest control [41]. The highest tyrosinase inhibition observed after treatment with the extracts from aerial organs of the studied species can be related with their higher content in phenolic compounds, since it is known that such molecules can significantly inhibit this enzyme [43].

### 2.5. Molecular Modeling

The inhibition assays performed on the extracts (Section 2.4) have shown a solid anti-tyrosinase activity. Among the substances identified by HPLC, and by comparison of the concentrations and their relative presence in the samples, we have selected four compounds, namely quercetin-di-Hex-dHex(**11**), quercetin-Hex-dHex (**12**), Kaempferol-Hex-Rut (**13**), and kaempferol-dHex-Hex (**14**), which have been subjected to molecular docking and prooved for their ability to interact with the tyrosinase enzymatic cavity (Figure 2 and Figure 3). The in silico experiments have shown that these substances are able to fit well into the enzymatic pocket of tyrosinase and bind to several residues. The presence of the sugar moieties on the four selected molecules enhances the capacity to interact with the copper atoms present in the catalytic cavity of the tyrosinase. On the other hand, the size of the substance could create a steric hindrance which does not allow to obtain a perfect insertion of the substances in the deep of the enzymatic pocket, considering the fact that the tyrosinase should recognize phenol-like substances.

Compounds **12**, **13**, and **14** are all able to interact with the Cu atoms of tyrosinase, thus these molecules are potentially able to strongly inhibit tyrosinase, whereas compound **11** has the lowest gold score (ChemScore), and the lowest calculated ΔG, probably due to the steric hindrance of the di-glucoside group present in the structure. Compound **13** has the best score (15.44) and the best ΔG (-25.98) and it is able to bind to Cu401, and to establish an H bond to Arg268 and stacks to His61 and His85, which are key residues of the enzyme. Compounds **12** and **14** are also able to bind respectively to Cu401 and Cu400, and to establish several hydrogen bonds and stacks; however, their scores and ΔG are lower than compound **13** probably due to steric factors (for detailed interactions see Table 5).

Taking the scores into consideration, the ΔG energy values and the ability to bind to the Cu atoms of tyrosinase, it can be hypothesized that compunds **12–14** may play a role in the anti-tyrosinase activity of the extracts. Also, it must be taken into consideration that the ∆G binding energy of the reference compounds, kojic acid and tropolone, are significantly higher (respectively −30.11 and −28.62 Kcal/mol) than that of compound **12–14**. Also, the chemscore values of the reference compounds are the higher of this series (Table 5). Moreover, considering that the correlation between the relative concentration of these substances and the anti-tyrosinase activity of the extracts is not linear, and that the docking experiments revealed medium–weak affinity for the tyrosinase enzyme, the total inhibitory activity found for the extracts might be influenced by the concomitant presence of other bioactive substances with synergistic or additive activity.

### 2.6. In Silico Absorption, Distribution, Metabolism, and Excretion (ADME) Evaluation

The ADME parameters of compounds **11**–**14** were predicted computationally, and results are depicted in Table 6. Acording to the obtained data, these molecules exhibit low oral absorption, no activity in the central nervous system (CNS), and, in general, unfavorably pharmacokinetic properties, as it could be expected from isolated flavonoids. In Section 2.4, it was observed that the studied extracts from sea rocket exhibited low to moderate activity towards enzymes related with the onset of neurodegenerative disorders, namely AChE and BuChE, and on T2DM, specifically α-glucosidase and α-amylase. This is supported by the ADME studies, that unravelled the poor pharmacokinetic properties of the main molecules present in such extracts, coupled with limited oral availability and nil activity in the CNS. Samples showed, however, strong inhibition towards tyrosinase, which is implicated in hyperpigmentation disorders, thus suggesting that they could be useful for external uses, such as for dermatological aplications. In fact, it is well known that phenolic glicosides, such as those identified in the sea rocket extracts, are well absorbed into skin dermis and have wide applications for cosmetic and dermatological purposes [44]. However, the final effect of orally administered extracts should also consider the biotransformations that may occur in the GI tract by enzymes and microbiota, thus, it is not completely possible to run out other possible systemic effects.

### 2.7. Antioxidant Activity

As a general rule, plants contain phenolic compounds with antioxidant properties, with potential applications as health promoters (e.g., to prevent oxidative stress-related diseases such as cardiovascular disorders and cancer), or as food additives (e.g., in food processing to improve the conservation processes and to extent shelf life). In this work, the extracts were appraised for in vitro antioxidant properties by five assays, and the results are summarized in Table 7.

Samples had, in general, nil to low antioxidant activity, and only the ethanol extract from the aerial organs had shown RSA towards the DPPH radical (IC_50_ = 0.59 mg/mL), and ferric reducing capacity (IC_50_ = 0.99 mg/mL). The higher activity detected in this extracts is likely due to its higher TIPC. Moreover, no relevant differences in antioxidant properties were observed between extracts from the different organs. The DPPH RSA of the aerial organs’ ethanol extracts from sea rocket presented in this work is in accordance with the values reported for methanol extracts from leaves of cultivated sea rocket plants in Tunisia, where IC_50_ values of at least 0.6 mg/mL were obtained, depending on the irrigation salinity [12]. However, a significant antiradical activity towards DPPH was detected in 80% methanol extracts from shoots of cultivated sea rocket plants, in Tunisia, in vegetative and flowering stages, with IC_50_ values ranging from 240 to 650 µg/mL (depending on harvesting period and irrigation salinity [13]). A high capacity to scavenge the DPPH radical was also observed for ethanol extracts from leaves, stems, and flowers of the same species [3]. Additionally, a significant activity on 2,2′-azino-bis(3-ethylbenzothiazoline-6-sulphonic acid (ABTS) (IC_50_ = 0.144 mg/mL) was observed in water-methanol extracts from sea rocket leaves, collected in Brittany (France) [14]. The differences between our results and other reports can be related with genetics and environmental factors (e.g., salinity), along with the tested extract, all factors known to significantly influence the chemical composition of plants, and therefore, its antioxidant potential.

## 3. Materials and Methods

### 3.1. Chemicals

Sigma-Aldrich (Hamburg, Germany) provided ABTS, DPPH, terc-butylhydroxytoluene (BHT), gallic acid, rutin, electric eel AChE (type-VI-S, EC 3.1.1.7), horse serum BuChE (EC 3.1.1.8), galanthamine, acetylthiocholine iodide (ATChI), butyrylthiocholine chloride (BTChI), 5,5-dithio-bis(2-nitrobenzoic) acid (DTNB), mushroom tyrosinase (EC1.14.18.1), glucosidase (3.2.1.20, from *Saccharomyces cerevisiae*), amylase (3. 2.1.1, from porcine pancreas), 4-nitrophenyl dodecanoate (NPD), N-Succinyl-Ala-Ala-Alap-nitroanilide (SANA), N-[3-(2-Furyl)acryloyl]-Leu-Gly-Pro-Ala (FALGPA), and 4- dimethylaminocinnamaldehyde (DMACA). Additional solvents and chemicals were provided by VWR International (Oud-Heverlee, Belgium), including those used in cell culture.

### 3.2. Plant Material

Sea rocket plants (voucher number XBH42) were collected in Faro Beach, South of Portugal (coordinates: 37°0′0.163″ N, −7°9′86.070″ W), in July of 2018. Plants were divided into aerial vegetative organs (leaves and stems) and mature fruits. Samples were cleaned from external contaminations with tap water, dried in an oven for 3 days at 50 °C, powdered, and stored at −20 °C until analysis.

### 3.3. Preparation of the Extracts

Extracts were prepared by an ultrasound-assisted extraction method using food grade solvents. In brief, dried biomass was mixed with ethanol, acetone, and water (1:40, *w*/*w*), and extracted for 30 min in an ultrasonic water bath, at room temperature (RT, circa 20 °C). Organic extracts were filtered (Whatman paper no. 4) to remove solid debris, and the solvent was completely removed under reduced pressure at 40 °C, in a rotary evaporator. Water extracts were freeze-dried. The obtained dried extracts were weighed, dissolved in methanol at the concentration of 50 mg/mL, and stored at −20 °C until analysis.

### 3.4. Toxicological Evaluation

#### 3.4.1. Cell Culture

To determine the in vitro toxic effects of the extracts, the following mammalian cell lines were used: RAW 264.7 macrophages (murine, provided by the Faculty of Pharmacy and Centre for Neurosciences and Cell Biology, University of Coimbra, Portugal), HEK 293 (human embryonic kidney, provided by the Functional Biochemistry and Proteomics group, Centre of Marine Sciences, Portugal), and HepG2 (human hepatocellular carcinoma, Marine Molecular Bioengineering groups, Centre of Marine Sciences, Portugal). RAW cells were maintained in Roswell Park Memorial Institute (RPMI) 1640 culture media, while HEK 293 and HepG2 cell lines were cultured in Dulbecco’s Modified Eagle Medium (DMEM) culture media, both supplemented with 10% heat-inactivated FBS, 1% L-glutamine (2 mM), and 1% penicillin (50 U/mL)/streptomycin (50 μg/mL), and were kept at 37 °C in a moistened atmosphere with 5% CO_2_.

#### 3.4.2. Citotoxicity

Cells were seeded in 96-well microplates in different densities, according to the cell line, as follows: RAW: 1 × 10^4^ cells/well, HEK and HepG2: 5 × 10^3^ cells/well. Cells were left to adhere for 24 h and treated with the extracts at the concentration of 100 µg/mL for 72 h. Control cells were treated with methanol at the highest concentration used in the treatments (0.2%). Cellular viability was determined by the 3-(4,5-dimethylthiazol-2-yl)-2,5-diphenyltetrazolium bromide (MTT) colourimetric assay [45]. Results were expressed in terms of cellular viability (%).

### 3.5. Identification and Quantification of Phenolic Compounds by HPLC-MS

The phenolic fraction of the extracts was characterized by resorting to an HPLC Agilent 1100 Series with a G1315B diode array detector. A Luna Omega Polar C_18_ analytical column of 150 × 3.0 mm and 5 µm particle size (Phenomenex), with a Polar C_18_ Security Guard cartridge (Phenomenex) of 4 × 3.0 mm, was used. The HPLC system was connected to an ion trap mass spectrometer (Esquire 6000, Bruker Daltonics) equipped with an electrospray interface operating in negative mode. Detailed conditions are reported elsewhere [46]. Compounds identification was carried out based on analytical standards and mass spectra, whereasultra violet (UV) spectra were used for quantification purposes.

### 3.6. Enzyme Inhibitory Activities

#### 3.6.1. Cholinesterase Inhibition

The cholinesterase inhibition potential of the extracts (0.5, 1, and 5 mg/mL) was evaluated by the Ellman’s method towards AChE and BuChE, as described in Zengin [47]. Galantamine (0.5–5 mg/mL) was used as the standard and results were expressed as the equivalent of galantamine (mg GALAE/g extract).

#### 3.6.2. Tyrosinase Inhibition

The tyrosinase inhibition assay was performed as described by Zengin [47], on the samples at concentrations between 0.5 and 5 mg/mL. Kojic acid was used as standard inhibitor (at the same concentration as the samples) and results were presented as the equivalent of kojic acid (mg KAE/g).

#### 3.6.3. Alpha-Amylase and α-Glucosidase Inhibition

The α-amylase and α-glucosidase inhibition capacity was evaluated according to the method reported by Uysal et al. [48]. Samples’ concentration were between 0.5 and 5 mg/mL and acarbose was used as standard (at the same concentration as the samples). Results were expressed as the equivalent of acarbose (mmol ACAE/g).

### 3.7. Molecular Modeling

#### 3.7.1. Receptor Preparation

The extracts reported in this paper have shown a considerable anti-tyrosinase activity. To study the possible interaction between selected molecules identified in the extracts and the tyrosinase catalytic pocket, a computational study by molecular docking was conducted. The in silico part of this work was oriented through the metalloenzyme tyrosinase. The crystallographic structure of mushroom tyrosinase (pdb: 2Y9X) [49] has been downloaded from the Protein Data Bank (PDB). Tropolone is co-crystallized in this structure, thus it has been used as a guide to delimitate the enzymatic cavity. The enzyme was cleaned as previously reported. Non-catalytic water was also removed, not being involved in the catalytic mechanism, and the enzyme was neutralized at pH 7.4 by the module PropKa of Maestro. All the errors of the crystal were fixed automatically be the PrepWizard tool embedded in Maestro 10.2 suite [50].

#### 3.7.2. Ligands Preparation

Considering the interesting anti-tyrosinase activity found for almost all the extracts, the effort was focused on four flavonoids, selected as the most representative polyphenols compounds, namely compound **11**: quercetin-3-*O*-di-glucoside-7-*O*-rhamnoside, compound **12**: quercetin-3-*O*-glucoside-7-*O*-rhamnoside, compound **13**: kaempferol-3-*O*-rutinoside-7-*O*-glucoside, and compound **14**: kaempferol-3-*O*-glucoside-7-O-rhamnoside. Also tropolone and kojic acid were downloaded from the zinc database and used for comparison and to validate the docking experiments. The chemical structures of compound **11–14** were drawn by ChemDraw 14 and then the two-dimensional (2D) structures were converted into three-dimensioanl (3D) by Maestro and the chirality was carefully checked. The module LigPrep was used to neutralize the molecules at pH 7.4 and the module ionizer was used to minimize the energy of the structure by applying the force field OPLS3 [51].

#### 3.7.3. ADME Estimation

After the preparation of the structures of compound **11–14** by the Schrodinger 2015-2 package, ADME of chemicals were computed by QikProp [52], implemented in the Schrodinger package as previopusly performed by our group [53]

#### 3.7.4. Self-Docking and Docking Method Validation

Following a previously reported paper by Stefanucci et al., the docking software Gold was tested for validation [54]. At this stage, the software GOLD 5.5 was configured for self-docking involving the crystallographic tyrosinase inhibition-enzyme complex (2Y9X), as described above. All the scoring functions of GOLD (ASP, PLP, GOLDSCORE, and CHEMSCORE) were considered to conduct the validation tests, performing a self-docking of the crystallographic ligands, and by comparing the root-mean-square deviation (RMSD) of the best docked crystallographic pose with the original crystallographic pose. An area of 15 Å around the co-crystallized ligand was defined as the binding site. At the end of the docking calculations, the ChemScore scoring function returned with the lowest RMSD value. Both ASP and PLP methods were not able to find any interactions for tropolone with the copper atomes, thus have been discarded. On the other hand, both Goldscore and Chemscore were able to find a similar pose of tropolone superimposable to the crystallographic one with similar RMSD in the range of 2–3 angstorms. The best RMSD value self-docking of tropolone was obtained by using the Chemscore methods (RMSD = 2.33 Å), which was considerate good following the data reported in other works e.g., References [55,56]. The self docking also revealed that the best pose is similar to the crystallographic pose of tropolone cocrystallized to the tyrosinase-like 1 enzyme [57].

Moreover, further docking experiments were performed on the chemical standard, used in this work to calculate the inhibitory potency, namely kojic acid. Kojic acid is indeed considered a powerful inhibitor of tyrosinase. The docking was performed in a similar manner as for tropolone and the G-binding and chemscore values obtained were compared to those of the other selected substances and are reported in Table 5. A similar study has been already conducted by Nokinsee et al. [58]. In this study, a comprehensive in silico predictive study on the tyrosinase inhibitors by comparing the binding data and the IC_50_ value for kojic acid and tropolone and other substances has been reported. It has been found that tropolone has a binding energy lower than that of kojic acid, also in agreement with the experimental IC_50_ values of tropolone (IC50 = 0.4 µM) and of kojic acid (IC_50_ = 7.4 µM). Our results are reported in Table 5 and are in full agreement with previously reported data [58]. After this validation process, the docking experiments were carried out also on compounds **11–14** and for a better understanding of the results, were compared with the results of tropolone and kojic acid.

#### 3.7.5. Molecular Docking

Dockings of the selected molecules were performed on the prepared tyrosinase structure by using the software Gold 5.5, developed by the Cambridge University [59]. ChemScore scoring function was employed for the docking calculations. The docking area was determined by centering the grid on the crystallographic inhibitor tropolone and extended in a radius of 15 Angstroms around the ligand center, similarly to the previous published works [60,61,62]. The best pose for each compound docked to tyrosinase was selected and depicted in Figure 2 and Figure 3. The binding energy and the chemscore values are reported in Table 5.

### 3.8. In Vitro Antioxidant Properties

Extracts were evaluated for in vitro antioxidant properties, at six different concentrations, ranging from 10 to 2000 µg/mL. Assays were performed in 96-well flat bottom microtitration plates, and absorbances were read on a multi-plate reader (EZ read 400, Biochrom) (Cambridge, UK). Results were expressed as a percentage of inhibition relative to the negative control containing methanol, and as IC_50_ values (mg/mL) whenever possible.

#### 3.8.1. Targeting Free Radicals: Radical Scavenging Activity (RSA) on DPPH and ABTS Radicals

The RSA on DPPH and ABTS radicals was performed as decribed previously [10]. The synthetic antioxidant compound BHT was used as positive control, in the same concentrations as the samples.

#### 3.8.2. Targeting Metal Ions: Metal Chelating Activity on Copper (CCA) and Iron (ICA) and Iron Reducing Power (FRAP)

CCA and ICA were determined as described elsewhere [63]. The synthetic metal chelator, ethylenediaminetetraacetic acid (EDTA), was used as the positive control, in the same concentrations as the extracts. Samples were also tested for FRAP [11], where an increase in samples’ absorbance represents an increase in the reducing power of the extracts. Results were expressed as percentage of inhibition relative to the standard (BHT) at the concentration of 1 mg/mL, and as IC_50_ (µg/mL) whenever possible.

### 3.9. Statistical Analyses

Statistical analysis was performed using Statistical Package for the Social Sciences (SPSS) Statistics software v.22 (IBM SPSS Statistics for Windows, IBM Corp., Chicago, USA). Data of all analyses, in triplicate, are expressed as mean ± standard error of the mean (SEM). A one-way analysis of variance (ANOVA) with Tukey’s honest significant difference (HSD) post-hoc test (*p* < 0.05) was used to look for statistically significant differences among results. Differences amongst samples were considered significant if *p* values were equal or inferior to 0.05. Half-maximal inhibitory concentration (IC_50_) values were determined through data sigmoidal fitting in the GraphPad Prism v. 5.0 software.

## 4. Conclusions

Ethanol, acetone, and water extracts from aerial organs and fruits of sea rocket were in general not toxic towards mammalian cell lines, and were mainly constituted by flavonoids, especially quercetin and kaempferol glycosides. The extracts displayed moderate and low capacity to inhibit AChE and BuChE, respectively. The extracts had reduced capacity to inhibit amylase and glucosidase, except for the ethanol fruits extracts, which could significantly inhibit the latter enzyme. ADME studies suggest that these molecules also exhibit low oral absorption. On the other hand, interestingly, except for the water extracts from fruits, tested samples had a strong inhibitory capacity towards tyrosinase containing quercetin and kaempferol glycosides that fit well into the enzymatic pocket of the latter enzyme. Samples had reduced antioxidant capacity. Our results suggest that the sea rocket can be further explored for its potential to provide pharmacological active natural products, especially with the capacity to inhibit tyrosinase, for cosmetic and dermatological purposes.

## Figures and Tables

**Figure 1 plants-09-00142-f001:**
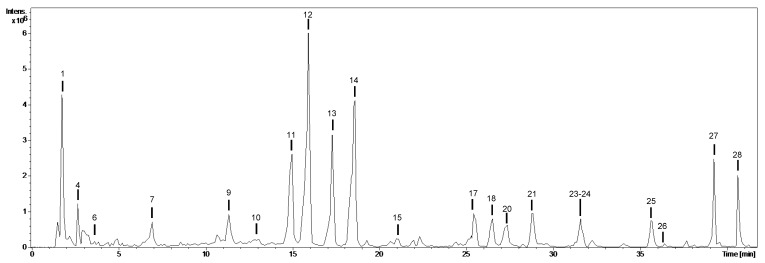
Base peak chromatogram of the water extract of sea rocket (*Cakile maritima*) fruits. Numbers refers to the compounds in Table 2 and Table 3.

**Figure 2 plants-09-00142-f002:**
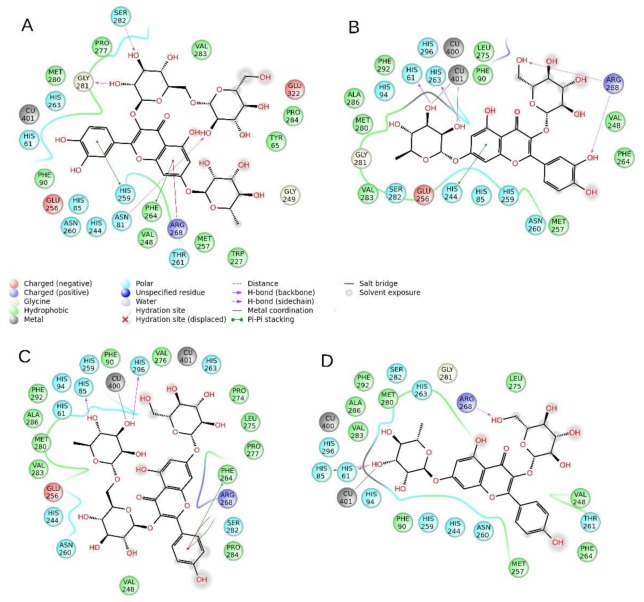
Interaction diagrams of compounds **11** (quercetin-di-Hex-dHex) (**A**), **12**: quercetin-Hex-dHex (**B**), **13**: Kaempferol-Hex-Rut (**C**), and **14**: kaempferol-dHex-Hex (**D**), identified in the sea rocket (*Cakile maritima*) extracts, docked to tyrosinase.

**Figure 3 plants-09-00142-f003:**
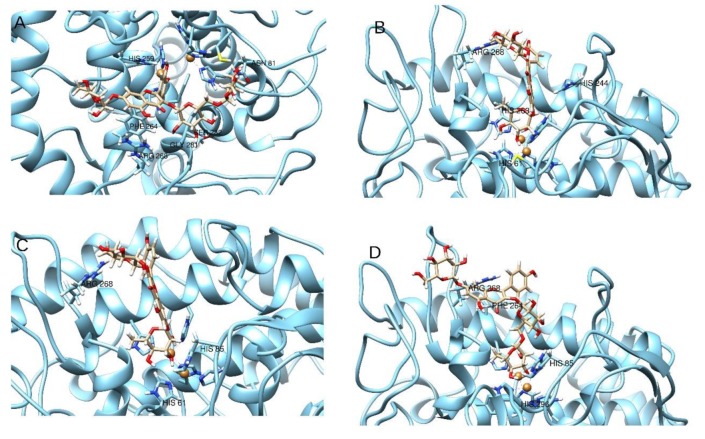
Three-dimensional (3D) best poses of compounds **11** (quercetin-di-Hex-dHex) (**A**), **12**: quercetin-Hex-dHex (**B**), **13**: Kaempferol-Hex-Rut (**C**), and **14**: kaempferol-dHex-Hex (**D**), identified in the sea rocket (*Cakile maritima*) extracts, docked to tyrosinase.

**Table 1 plants-09-00142-t001:** Effect of the application of 100 µg/mL of food grade extracts prepared from aerial vegetative organs and fruits of sea rocket (*Cakile maritima*) during 72 h on the viability of RAW 264.7 (murine macrophages), HEK 293 (human embryonic kidney), and HepG2 (human hepatocellular carcinoma) cell lines.

Organs	Extract	RAW 264.7	HEK 293	HepG2
Aerial vegetative organs	Ethanol	67.7 ± 4.50 ^b^	78.4 ± 6.01 ^b^	103 ± 9.12 ^a^
	Acetone	83.1 ± 7.91 ^a^	72.2 ± 2.72 ^b^	78.3 ± 19.1 ^b^
	Water	72.3 ± 1.52 ^ab^	76.5 ± 15.0 ^b^	115 ± 13.5 ^a^
Fruits	Ethanol	80.9 ± 4.80 ^a^	82.2 ± 5.33 ^b^	101 ± 16.7 ^a^
	Acetone	73.1 ± 12.9 ^a^	79.4 ± 4.70 ^b^	96.2 ± 6.52 ^a^
	Water	90.1 ± 14.6 ^a^	140 ± 25.1 ^a^	110 ± 4.02 ^a^

Results are expressed as the mean of the % of viability relative to a control containing methanol (0.2%, *v*/*v*) ± standard error of mean (SEM), n = 6. For the same column, different letters are significantly different (Multiple Comparisons of Means: Tukey Contrast, 95% family-wise confidence level).

**Table 2 plants-09-00142-t002:** Qualitative profile of the compounds present in extracts from aerial vegetative organs and fruits of sea rocket (*Cakile maritima*).

No.	t*_R_* (min)	[M-H]^−^ *m*/*z*	m/z (% Base Peak)	Assigned Identification	Fruits			Aerial Organs		
					Water	Ethanol	Acetone	Water	Ethanol	Acetone
1	1.8	377	MS^2^ [377]: 341 (100) MS^3^ [377→341]: 179 (64), 161 (59), 143 (100), 119 (5), 113 (7)	Disaccharide (HCl adduct)	√	√		√		√
2	2.0	374	MS^2^ [374]: 294 (16), 275 (68), 259 (100), 241 (14), 227 (16), 163 (17), 145 (28)	Dihydrogluconapin isomer-1		√	√	√	√	√
3	2.6	360	MS^2^ [360]: 275 (100), 259 (86), 257 (25), 119 (71)	Unknown			√		√	√
4	2.7	191	MS^2^ [191]: 173 (37). 111 (100)	Citric acid	√			√		
5	3.1	374	MS^2^ [374]: 294 (9), 275 (14), 259 (100), 241 (12), 163 (12), 145 (6)	Dihydrogluconapin isomer-2		√	√	√	√	√
6	3.7	315	MS^2^ [315]: 153 (100) MS^3^ [315→153]: 109 (100)	Dihydroxybenzoic acid hexoside	√			√		
7	7.0	323	MS^2^ [323]: 179 (74), 161 (53), 119 (99), 113 (100), 101 (30)	Hexose derivative	√	√				
8	10.7	431	MS^2^ [431]: 385 (100) MS^3^ [431→385]: 223 (100), 205 (51), 161 (34), 153 (54)	Roseoside (formate adduct)				√	√	√
9	11.4	294	MS^2^ [294]: 279 (100), 264 (10)	Unknown	√	√	√			
10	13.1	609	MS^2^ [609]: 447 (100), 301 (54) MS^3^ [609→447]: 301 (100) MS^4^ [609→447→301]: 271 (100), 255 (54), 179 (15)	Quercetin-*O*-hexoside-*O*-deoxyhexoside	√		√			
11	15.0	771	MS^2^ [771]: 609 (55), 463 (100), 301 (28) MS^3^ [771→463]: 343 (46), 301 (100), 271 (35), 179 (18), 151 (21)	Quercetin-*O*-dihexoside-*O*-deoxyhexoside	√	√	√	√	√	√
12	15.9	609	MS^2^ [609]: 463 (33), 447 (100), 301 (50) MS^3^ [609→447]: 301 (100) MS^4^ [609→447→301]: 301 (100), 179 (22), 151 (8)	Quercetin-*O*-hexoside-*O*-deoxyhexoside	√	√	√	√	√	√
13	17.3	755	MS^2^ [755]: 593 (100), 447 (76), 285 (43) MS^3^ [755→593]: 285 (100), 255 (42)	Kaempferol-*O*-hexoside-*O*-rutinoside	√	√	√	√	√	√
14	18.6	593	MS^2^ [593]: 447 (100), 431 (41), 285 (42) MS^3^ [593→447]: 285 (36), 284 (100), 255 (42), 151 (12)	Kaempferol-*O*-deoxyhexoside-*O*-hexoside	√	√	√	√	√	√
15	21.0	463	MS^2^ [463]: 301 (100) MS^3^ [463→301]: 271 (47), 179 (100), 151 (74)	Quercetin-*O*-hexoside	√	√		√	√	
16	23.9	447	MS^2^ [447]: 285 (100), 255 (16) MS^3^ [447→285]: 255 (100), 227 (12)	Kaempferol-*O*-hexoside				√		
17	25.4	580	MS^2^ [580]: 580 (100), 373 (23), 223 (15)	Unknown	√	√	√	√	√	
18	26.5	609	MS^2^ [609]: 301 (100) MS^3^ [609→301]: 179 (14), 151 (100)	Quercetin-*O*-rutinoside	√	√	√	√	√	√
19	26.5	753	MS^2^ [753]: 529 (100) MS^3^ [753→529]: 511 (52), 247 (78), 223 (100) MS^4^ [753→529→223]: 208 (100)	Disinapoylgentiobioside			√	√	√	√
20	27.3	539	MS^2^ [539]: 377 (100), 307 (63), 275 (58) MS^3^ [539→377]: 307 (70), 275 (100)	Oleuropein	√	√				
21	28.8	447	MS^2^ [447]: 301 (100) MS^3^ [447→301]: 179 (18), 151 (100)	Quercetin-*O*-deoxyhexoside	√	√	√	√	√	√
22	30.3	591	MS^2^ [591]: 367 (86), 223 (100) MS^3^ [591→223]: 208 (69), 164 (100)	Disinapoyl-hexoside				√	√	√
23	31.2	959	MS^2^ [959]: 735 (100), 529 (11) MS^3^ [959→735]: 717 (14), 529 (100), 511 (76), 497 (43), 457 (18) MS^4^ [959→735→529]: 245 (100), 223 (73)	Trisinapoylgentiobioside		√	√			
24	31.5	593	MS^2^ [593]: 285 (100) MS^3^ [593→285]: 241 (100), 151 (50)	Kaempferol-*O*-rutinoside	√	√	√	√	√	√
25	35.6	431	MS^2^ [431]: 285 (100) MS^3^ [431→285]: 257 (36), 151 (100)	Kaempferol-*O*-deoxyhexoside	√	√	√	√	√	√
26	36.3	461	MS^2^ [461]: 315 (100), 300 (19)	Isorhamnetin-*O*-deoxyhexoside	√	√	√			
27	39.2	327	MS^2^ [327]: 291 (35), 229 (100), 211 (84), 171 (62)	Oxo-dihydroxy-octadecenoic acid	√	√	√	√	√	√
28	40.5	329	MS^2^ [329]: 311 (33), 229 (100), 211 (66), 171 (77)	Trihydroxy-octadecenoic acid	√	√	√	√	√	√

**No.: number of the peak; t*_R_* (min): retention time; [M-H]^−^: deprotonated molecular ion; m/z: mass to charge ratio; MS: mass spectra.**

**Table 3 plants-09-00142-t003:** Quantitative profile (mg/gdry extract) of the main phenolic compounds present in extracts prepared from aerial vegetative organs and fruits of sea rocket (*Cakile maritima*).

		Fruits			Aerial Organs		
Compounds *	Assigned Identification	Water	Ethanol	Acetone	Water	Ethanol	Acetone
*Flavonoids*							
11	Quercetin-di-Hex-dHex	1.42 ± 0.06 ^a^	1.20 ± 0.06 ^b^	0.16 ± 0.01 ^d^	1.05 ± 0.05 ^c^	1.25 ± 0.07 ^b^	0.12 ± 0.008 ^d^
12	Quercetin-Hex-dHex	1.30 ± 0.07 ^b^	1.05 ± 0.06 ^c^	0.43 ± 0.03 ^e^	0.86 ± 0.04 ^d^	1.56 ± 0.08 ^a^	0.48 ± 0.03 ^e^
13	Kaempferol-Hex-Rut	1.14 ± 0.06 ^c^	0.95 ± 0.04 ^cd^	0.26 ± 0.01 ^e^	2.80 ± 0.10 ^b^	4.20 ± 0.30 ^a^	0.70 ± 0.04 ^d^
14	Kaempferol-dHex-Hex	0.73 ± 0.04 ^d^	0.49 ± 0.03 ^de^	0.29 ± 0.02 ^e^	1.80 ± 0.10 ^b^	3.60 ± 0.20 ^a^	1.50 ± 0.10 ^c^
16	Kaempferol-Hex	nd	nd	nd	0.11 ± 0.01	nd	nd
18	Quercetin-Rut	0.18 ± 0.01	0.06 ± 0.00	nd	nd	nd	nd
21	Quercetin-dHex	0.15 ± 0.01 ^a^	0.11 ± 0.01 ^b^	0.05 ± 0.00 ^c^	0.04 ± 0.00 ^c^	0.11 ± 0.01 ^b^	0.02 ± 0.00 ^c^
24	Kaempferol-Rut	0.13 ± 0.01 ^d^	0.03 ± 0.00 ^e^	0.0029 ± 0.0002 ^e^	0.85 ± 0.04 ^a^	0.23 ± 0.01 ^c^	0.54 ± 0.03 ^b^
25	Kaempferol-dHex	0.091 ± 0.005 ^cd^	0.081 ± 0.003 ^d^	0.08 ± 0.00 ^d^	0.12 ± 0.01 ^c^	0.37 ± 0.02 ^a^	0.23 ± 0.01 ^b^
26	Isorhamnetin-dHex	---	---	0.04 ± 0.00	---	---	---
Total		5.10 ± 0.10 ^c^	4.00 ± 0.10 ^d^	1.28 ± 0.04 ^e^	7.6 ± 0.2 ^b^	11.3 ± 0.4 ^a^	3.6 ± 0.1 ^d^
*Others*							
20	Oleuropein	0.26 ± 0.01	0.27 ± 0.01	nd	nd	nd	nd
22	Disinapoyl-Hex	0.12 ± 0.01 ^c^	0.15 ± 0.01 ^bc^	nd	0.26 ± 0.02 ^a^	0.17 ± 0.01 ^b^	0.17 ± 0.01 ^b^
Total		0.38 ± 0.01 ^b^	0.42 ± 0.01 ^a^	nd	0.26 ± 0.02 ^c^	0.17 ± 0.01 ^d^	0.17 ± 0.01 ^d^
TIPC		5.48 ± 0.1 ^c^	4.42 ± 0.10 ^d^	1.28 ± 0.04 ^f^	7.92 ± 0.2 ^b^	11.5 ± 0.4 ^a^	3.94 ± 0.11 ^e^

* Compounds refer to those presented in Table 1 and Figure 1. TIPC: Total individual phenolic content (the sum of all phenolics individually quantified). For the same row, different letters are significantly different (Multiple Comparisons of Means: Tukey Contrast, 95% family-wise confidence level).

**Table 4 plants-09-00142-t004:** Enzymatic inhibitory activity on acetyl- (AChE), butyrylcholinesterase (BuChE), α-glucosidase, α-amilase and tyrosinase, of different extracts prepared from aerial vegetative organs and fruits of sea rocket (*Cakile maritima*) *. Results are expressed as IC_50_ values (mg/mL).

Organs	Extract	AChE (mg GALAE/g)	BuChE (mg GALAE/g)	Amylase (mmol ACAE/g)	Glucosidase (mmol ACAE/g)	Tyrosinase (mg KAE/g)
Aerial organs	Ethanol	1.33 ± 0.14 ^a^	0.58 ± 0.10 ^b^	0.18 ± 0.01 ^a^	na	25.9 ± 0.13 ^a^
	Acetone	1.35 ± 0.01 ^a^	0.66 ± 0.19 ^b^	0.26 ± 0.01 ^a^	na	24.7 ± 0.13 ^a^
	Water	0.53 ± 0.03 ^b^	0.06 ± 0.01 ^d^	0.02 ± 0.01 ^b^	0.16 ± 0.01 ^b^	19.9 ± 0.12 ^b^
Fruits	Ethanol	1.33 ± 0.11 ^a^	0.86 ± 0.09 ^a^	0.20 ± 0.01 ^a^	2.19 ± 0.02 ^a^	24.9 ± 0.25 ^a^
	Acetone	1.29 ± 0.01 ^a^	0.92 ± 0.07 ^a^	0.21 ± 0.02 ^a^	0.16 ± 0.04 ^b^	24.0 ± 0.33 ^a^
	Water	1.26 ± 0.06 ^a^	0.26 ± 0.05 ^c^	0.04 ± 0.01 ^b^	0.24 ± 0.04 ^b^	6.16 ± 0.30 ^c^

* Values are expressed are means ± standard error of mean (SEM) of three parallel measurements. For the same column, different letters are significantly different (Multiple Comparisons of Means: Tukey Contrast, 95% family-wise confidence level). IC_50_: half maximal inhibitory concentration; GALAE: Galatamine equivalent; KAE: Kojic acid equivalent; ACAE: Acarbose equivalent; na: not active.

**Table 5 plants-09-00142-t005:** Summary of the interactions found in the best pose of Kojic acid, tropolone, and compounds **11**–**14** identified in the sea rocket (*Cakile maritima*) extracts, docked to tyrosinase.

Compounds	Residues											Copper Atoms		∆G	Chem Score
	His61	Asn81	His85	His244	His259	His263	Phe264	Arg268	Gly281	Ser282	His296	Cu400	Cu401		
**11**	--	H-bond	--	--	π–π	--	π–π	Cat-π	H-bond	H-bond	--	--	--	−8.87	1.29
**12**	π–π	--	--	π–π	--	H-bond	--	2× H-bonds	--		--	--	Coordinative bond	−26.46	13.22
**13**	--	--	H-bond	--	--	--	π–π	Cat-π	--	--	H-bond	Coordinative bond	--	−24.33	5.16
**14**	H-bond	--	H-bond	--	--	--	--	H-bond	--	--	--	--	Coordinative bond	−25.98	15.44
**Tropolone**	π–π	--	--	--	--	--	--	--	--	--	--	--	Coordinative bond	−28.62	26.60
**Kojic acid**	H-bond	H-bond To Asn 260	--	--	--	--	--	--	--	--	--	Coordinative bond	Coordinative bond	−30.11	25.43

**11**: quercetin-di-Hex-dHex; **12**: quercetin-Hex-dHex; **13**: Kaempferol-Hex-Rut; **14**: kaempferol-dHex-Hex.

**Table 6 plants-09-00142-t006:** The computed ADME parameters for the compounds **11**–**14** identified in the sea rocket (*Cakile maritima*) extracts.

Ligand	QPlogP O/W ^a^	QPPCaco ^b^	QPLogBB ^c^	Qual. Model for Human Oral Absorption ^d^	Lipinski Rule of 5 Violations ^e^	Jorgensen Rule of 3 Violations ^f^	CNS ^g^	HERG K^+ h^
**11**	−1.944	1.49	−4.788	low	3	2	−−	−6.198
**12**	−4.130	0.094	−7.230	low	3	2	−−	−6.858
**13**	−2.448	0.665	−5.412	low	3	2	−−	−6.298
**14**	−4.9	0.032	−7.856	low	3	2	−−	−6.590

**11**: quercetin-di-Hex-dHex; **12**: quercetin-Hex-dHex; **13**: Kaempferol-Hex-Rut; **14**: kaempferol-dHex-Hex. **ADME:** absorption, distribution, metabolism, and excretion studies; ^a^ Predicted Octanol/Water partition coefficient (reasonable value from (−2.0 to 6.5); ^b^ Predicted apparent Caco-2 cell permeability in nm/s (<25 poor, >500 great); ^c^ Predicted brain/blood partition coefficient (reasonable value from −3.0 to 1.2);^d^ Qual. Model for Human Oral Absorption (>80 high); ^e^ Lipinski Rule of 5 Violations (maximum is 4); ^f^ Jorgensen Rule of 3 Violations (maximum is 3); ^g^ Predicted CNS Activity (−− to ++); ^h^ HERG K^+^ Channel Blockage: log IC_50_ (concern below −5).

**Table 7 plants-09-00142-t007:** Radical scavenging on 2,2-diphenyl-1-picrylhydrazyl (DPPH) and 2,2’-azino-bis(3-ethylbenzothiazoline-6-sulfonic acid (ABTS) radicals, ferric reducing antioxidant power (FRAP), copper chelating (CCA), and iron chelating activities (ICA) of different extracts prepared from aerial vegetative organs and fruits of sea rocket (*Cakile maritima*). Results are expressed as IC_50_ values (mg/mL).

Organs	Extract	DPPH	ABTS	FRAP	CCA	ICA
Aerial organs	Ethanol	0.59 ± 0.35 ^b^	5.14 ± 0.33 ^b^	0.99 ± 0.05 ^a^ *	8.53 ± 0.31 ^c^	nr
	Acetone	nr	nr	1.38 ± 0.09 ^a^ *	nr	nr
	Water	nr	5.69 ± 2.30 ^c^	1.12 ± 0.08 ^a^	2.80 ± 0.30 ^b^	nr
Fruits	Ethanol	nr	4.88 ± 0.79 ^b^	2.03 ± 0.15 ^b^	8.17 ± 0.65 ^c^	nr
	Acetone	nr	nr	2.80 ± 0.48 ^b^	nr	nr
	Water	nr	6.74 ± 0.95 ^c^	1.31 ± 0.32 ^a^	2.53 ± 0.13 ^b^	5.23 ± 0.41 ^b^
BHT *		0.1 ± 0.02 ^a^	0.06 ± 0.0 ^a^			
EDTA *					0.11 ± 0.00 ^a^	0.07 ± 0.00 ^a^

Values represent the mean ± standard error of mean (SEM) performed six times (n = 6). For the same column, different letters are significantly different (Multiple Comparisons of Means: Tukey Contrast, 95% family-wise confidence level). nr: not reached; IC_50_: half maximal inhibitory concentration; * positive control tested at 1 mg/mL.

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
