# Peer review of "Phenolic Profile, Toxicity, Enzyme Inhibition, In Silico Studies, and Antioxidant Properties of Cakile maritima Scop. (Brassicaceae) from Southern Portugal"

_plants, 2020, doi:10.3390/plants9020142_

Round 1

Reviewer 1 Report

Dear Authors,

Reviewer comments plants-672409

The manuscript “Phenolic profile, toxicity, enzyme inhibition, in silico studies and antioxidant properties of Cakile maritima Scop. (Brassicaceae) from Souther Portugal” represents an edible halophyte with several medicinal uses.

In general, the manuscript is well written and presented. However, I am still not clearly understanding which compounds in the article are described for the first time in the species?

Some comments are provided below regarding some suggeating about jow to improve the manuscripts.

Other comments

Abstract:

This manuscript is interesting about the medicinal function of halophyte. However,

in the abstract section, authors have written that “some compounds are here described for the first time in the species”, could you explain which compounds? Moreover, “Samples were more active towards AChE, than on BuChE.” Could you explain the samples from which extract, all the samples from fruit and aerial organ extract, or from fruit extract, or from aerial extract? line 24, it should be “BuChE”

Keywords:

Line 34: the letter size of “bioactive plant-derived products; tyrosinase inhibitors” should be same with the letter of “salt tolerant plants”.

Introduction:

Line 39-43: The letter size from” represent about 2% of terrestrial plant species and are present in about half the higher plant families. They exhibit a high diversity of plant forms and are currently considered an important reservoir of bioactive molecules with multiple biotechnological applications, ranging from food to cosmetic ingredients [1-3]. There are reports of the traditional medicinal uses of 43 families, comprising more than 180 halophytic species, in the Mediterranean, the Arabian Sea and Syria regions” should be change.

Line 44-46: The letter size from” the treatment of pain, fever, liver and digestive disorders, skin, respiratory, genitourinary conditions, microbial and parasitic infections, inflammation, dermatitis, wounds, and burns [1, 4, 5].” should be change.

Line72: it should be “BuChE”

Results and Discussion:

Line 83-84: The letter size from” Despite the theoretical concept that natural products are safer than synthetic counterparts, it is well known that p” should be change.

Line 85-87: the letter size from “to assure its safety and potential interest to be further explored for biotechnological uses. Different methods are used for this purpose, including the in vitro assessment of acute toxic effects on mammallian cell lines.” should be change. In addition, should be “. Different methods, including the in vitro assessment of acute toxic effects on mammallian cell line, are used for this purpose.”

Line 98: remove “and “ after therefore.

Line 186-187: could explain why “For both compounds, their abundance in fruits was in the order acetone < ethanol< water, while in the aerial organs the order was acetone < water < ethanol”

Line 188-191: In table 3, the total individual phenolic content was defined as the sum of all the individual compounds. However, it looks like the results are not correct, like fruit water extract should be 5.14…. fruit ethanol extract should be 3.97…. May round up and round down numbers when calculated the individual compounds. Could check and rewrite them again. Finally, should change the value of TIPC, too.

Line 192: should be “2.4 Enzyme inhibitory activities”.

Line 204-210: should be “BuChE” in Table 4.

Line 247: should be “2.5 Molecular modelling

Line 248: should be “subsection 2.4”

Line 265: should be “Table 5”

Line 267-268: the number of the table should be “Table 5”

Line 274: should be “2.6 Antioxidant activity

Line 279: should be “Table 6”

Materials and methods:

In subsection 3.2 Plant material, the samples were dried in oven for 3 days at 50oC, is it enough for make powder? Moreover, is it possible to reduce compounds composition if use this method to dry samples.

References:

Line 445: the year should be before the page of the journal.

Line 446: should add the book name, editors, publisher, page….

The year from the journal has to be in bold in all references, like line 446, line 449.

Line 458: should delete “10.” before “Fuochi,V.”

Line 462: should delete “11” before “Azim, W.M.”

Line 469: should change “08” to “8”

Line 492: should add full name of the page, change “6611-9” to “6611-6619”, same in line 510, line 530, line 561, line 568.

Line 520: should add “Second Edition” and publisher name before “RSM Press

Author Response

Comment #1: In general, the manuscript is well written and presented. However, I am still not clearly understanding which compounds in the article are described for the first time in the species? Answer: The compounds are mentioned in lines 154 and 157: roseoside and oleuropein, respectively. This information was included in the abstract.

Comment #2: In the abstract section, authors have written that “some compounds are here described for the first time in the species”, could you explain which compounds? Answer: The compounds are mentioned in lines 154 and 157: roseoside and oleuropein, respectively. This information was included in the abstract.

Comment #3: Moreover, “Samples were more active towards AChE, than on BuChE.” Could you explain the samples from which extract, all the samples from fruit and aerial organ extract, or from fruit extract, or from aerial extract? line 24, it should be “BuChE” Answer: All samples were more active towards AChE. The sentence was modified in order to better reflect this. BuChe was corrected.

Comment #4: Line 34: the letter size of “bioactive plant-derived products; tyrosinase inhibitors” should be same with the letter of “salt tolerant plants”. Answer: This was corrected.

Comment #5: Line 39-43: The letter size from” represent about 2% of terrestrial plant species and are present in about half the higher plant families. They exhibit a high diversity of plant forms and are currently considered an important reservoir of bioactive molecules with multiple biotechnological applications, ranging from food to cosmetic ingredients [1-3]. There are reports of the traditional medicinal uses of 43 families, comprising more than 180 halophytic species, in the Mediterranean, the Arabian Sea and Syria regions” should be change. Answer: This was corrected.

Comment #6: Line 44-46: The letter size from” the treatment of pain, fever, liver and digestive disorders, skin, respiratory, genitourinary conditions, microbial and parasitic infections, inflammation, dermatitis, wounds, and burns [1, 4, 5].” should be change. Answer: This was corrected.

Comment #7: Line72: it should be “BuChE” Answer: This was corrected.

Comment #8: Line 83-84: The letter size from” Despite the theoretical concept that natural products are safer than synthetic counterparts, it is well known that p” should be change. Answer: This was corrected.

Comment #9: Line 85-87: the letter size from “to assure its safety and potential interest to be further explored for biotechnological uses. Different methods are used for this purpose, including the in vitro assessment of acute toxic effects on mammallian cell lines.” should be change. In addition, should be “. Different methods, including the in vitro assessment of acute toxic effects on mammallian cell line, are used for this purpose.” Answer: This was corrected.

Comment #10: Line 98: remove “and “ after therefore. Answer: This was corrected.

Comment #11: Line 186-187: could explain why “For both compounds, their abundance in fruits was in the order acetone < ethanol< water, while in the aerial organs the order was acetone < water < ethanol” Answer: There are not significant differences for all compounds. Hence, the sentence was changed to ‘the highest concentrations were found in ethanol and water, whereas acetone extracts presented the lowest concentrations in all cases’.

Comment #12: Line 188-191: In table 3, the total individual phenolic content was defined as the sum of all the individual compounds. However, it looks like the results are not correct, like fruit water extract should be 5.14…. fruit ethanol extract should be 3.97…. May round up and round down numbers when calculated the individual compounds. Could check and rewrite them again. Finally, should change the value of TIPC, too. Answer: Results were rounded considering the standard deviation (1 significant figure). Therefore, 5.14 was rounded to 5.1± 0.1. However, some results have been corrected in the revised version, rounding them accordingly.

Comment #13: Line 192: should be “2.4 Enzyme inhibitory activities”. Answer: This was corrected.

Comment #14: Line 204-210: should be “BuChE” in Table 4. Answer: This was corrected.

Comment #15: Line 247: should be “2.5 Molecular modelling” Answer: This was corrected.

Comment #16: Line 248: should be “subsection 2.4” Answer: This was corrected.

Comment #17: Line 265: should be “Table 5” Answer: This was corrected.

Comment #18: Line 267-268: the number of the table should be “Table 5” Answer: This was corrected.

Comment #19: Line 274: should be “2.6 Antioxidant activity” Answer: This was corrected.

Comment #20: Line 279: should be “Table 6” Answer: This was corrected.

Comment #21: In subsection 3.2 Plant material, the samples were dried in oven for 3 days at 50oC, is it enough for make powder? Moreover, is it possible to reduce compounds composition if use this method to dry samples. Answer: In this species it is possible to obtain dry biomass after 3 days at that temperature. It is known that the drying process affects the chemical composition of the biomass, and as a consequence, the chemical components of the obtained extracts.The specific influence could only be appraised by testing different drying methods on the metabolomics of obtained extracts, which was not the goal of this work.

Comment #22: Line 445: the year should be before the page of the journal.

Comment #23: Line 446: should add the book name, editors, publisher, page….

Comment #24: The year from the journal has to be in bold in all references, like line 446, line 449.

Comment #25: Line 458: should delete “10.” before “Fuochi,V.”

Comment #26: Line 462: should delete “11” before “Azim, W.M.”

Comment #27: Line 469: should change “08” to “8”

Comment #28: Line 492: should add full name of the page, change “6611-9” to “6611-6619”, same in line 510, line 530, line 561, line 568.

Comment #29: Line 520: should add “Second Edition” and publisher name before “RSM Press

Answer: The reference list was revised and corrected according to the reviewers remarks.

Reviewer 2 Report

Reviewed manuscprit is interesting due to the kind of the raw material (halophyte - sea rocket) and a very extensive and complementary experiment (detailed analysis of the composition of fruits and aerial vegetative organs extracts and their various biological properties). Only small correction is needed. In my opinion, the discussion part lacked explanation of the observed differences in the activity of extracts, e.g. against glucosidase and tyrosinase (lines 221-226 and 233-236) and HEK 293 and HepG2 cells (lines 91-100). Please compare quantitative and qualitative differences of extracts for extreme variants.  

Author Response

 Reviewer #2

Reviewed manuscprit is interesting due to the kind of the raw material (halophyte - sea rocket) and a very extensive and complementary experiment (detailed analysis of the composition of fruits and aerial vegetative organs extracts and their various biological properties).

Comment #1. Only small correction is needed. In my opinion, the discussion part lacked explanation of the observed differences in the activity of extracts, e.g. against glucosidase and tyrosinase (lines 221-226 and 233-236) and HEK 293 and HepG2 cells (lines 91-100). Please compare quantitative and qualitative differences of extracts for extreme variants. Answer. The discussion part was improved, as suggested by the reviewer.

Reviewer 3 Report

The manuscript entitled "Phenolic profile, toxicity, enzyme inhibition, in silico studies and antioxidant properties of Cakile maritima Scop. (Brassicaceae) from Southern Portugal" describes an interesting work focused on a few investigated species, worth to be better tested. 

Authors share their expertise and produced a novel work that mixed chemical analyses, biological essays and docking study.

The manuscript is well organized.

Nevertheless, one important major pitfall should be considered.

The in silico study can not be discussed in the current version since authors did not prove the reliability of computational retrieves replicating enzymatic assay on single compounds. This is a mandatory point, as well as to perform docking analysis applied to kojic acid, the reference drug. Also the discussion of in silico results should be deeply revised in order to give scientific soundness to the whole work. 

Round 2

Reviewer 3 Report

Authors improved the in silico section and ameliorated the results presentation.

Some important pitfalls remain:

1- Could authors prove the in silico analysis replicating anti-tyrosinase activity with single compounds 11-14? It would be valuable and very important to confirm the docking study that is only previsional without an enzymatic experimental test.

2- discussion and conclusions are too general and authors have to really better describe the different activity of extracts tested on different biological targets in order to better address limits and possibilities of application of the species for medicinal and health purposes.

I want to suggest to authors to perform at least an in silico ADME simulation or discuss the published pharmacokinetic of phenolics derivatives if they want to discuss the possible role of the constituents in brain disorders and at systemic level. This could be, in my opinion, a well-used 1 week work and it'd be an actual improvement of the whole work since it could open a wide and helpful discussion regarding this few investigated species.

The easiest way, on the other hand, is to deeply revise and enlarge the discussion of results focusing the attention on the different activity on targets, i.e. the better activity on tyrosinase. Indeed, the scarce activity on amilase or glucosidase and an almost good but not excellent activity on AChE are already "a first pharmacokinetic confirm" because the glicosides of the extracts can't be absorbed unmodified at systemic level and, as I mentioned before, we don't know if they could pass EEB. The best activity on tyrosinase is easier to discuss and it is well known that phenolic glicosides are well absorbed into skin dermis and they are interesting for cosmetic and dermatological purposes.

Minor issues: please revise the whole manuscript since many spelling errors occur; check subscripts.

Author Response

Reviewer #3

Comment #1. Could authors prove the in silico analysis replicating anti-tyrosinase activity with single compounds 11-14? It would be valuable and very important to confirm the docking study that is only previsional without an enzymatic experimental test.

Response: Dear Referee, thank you for your kind suggestion, however compounds 11-14 are not available at this moment as pure substances, thus it is not possible to perform further investigation.

Comment #2. Discussion and conclusions are too general and authors have to really better describe the different activity of extracts tested on different biological targets in order to better address limits and possibilities of application of the species for medicinal and health purposes. I want to suggest to authors to perform at least an in silico ADME simulation or discuss the published pharmacokinetic of phenolics derivatives if they want to discuss the possible role of the constituents in brain disorders and at systemic level. This could be, in my opinion, a well-used 1 week work and it'd be an actual improvement of the whole work since it could open a wide and helpful discussion regarding this few investigated species.

The easiest way, on the other hand, is to deeply revise and enlarge the discussion of results focusing the attention on the different activity on targets, i.e. the better activity on tyrosinase. Indeed, the scarce activity on amilase or glucosidase and an almost good but not excellent activity on AChE are already "a first pharmacokinetic confirm" because the glicosides of the extracts can't be absorbed unmodified at systemic level and, as I mentioned before, we don't know if they could pass EEB. The best activity on tyrosinase is easier to discuss and it is well known that phenolic glicosides are well absorbed into skin dermis and they are interesting for cosmetic and dermatological purposes.

Response: We are grateful to the reviewer for these valuable comments. We performed in silico ADME on compounds 11-14. The data obtained are reported in Table 6 of the paper. As expected they have very poor pharmacokinetic properties, limited oral availability and no activity in the central nervous system system (CNS). Thus, the extracts of this specific plants could be useful for external uses such as dermatological uses. However, the final effect of extract administered orally should keep in consideration the biotransformation that will occur in the gastrointestinal tract by enzymes and microbiota, thus, it is not completely true to run out other possible systemic effects. In view of ADME results, the discussion section was improved, and the conclusions were improved.

Comment #4. Minor issues: please revise the whole manuscript since many spelling errors occur; check subscripts.

Response: The whole manuscript was revised, and errors and other typos were corrected.

Round 3

Reviewer 3 Report

I read the revised version of the manuscript. 

I find that authors addresses all major and minor concerns as per reviewers' comments.

Some suggestions:

194: extracts

234: latter (the second mentioned before) or later (metabolites)?

256: ... organs of the studied species

284-285: ... may play a role in the anti-tyrosinase activity of the extracts

285: check special characters

300: specifically

301: delete: This suggests that the extracts may not serve as targets for further studies aimimg its use in therapeutics targeting such health disorders.

306: before In fact insert: . and correct applications

458: delete D. 

From 499: just a suggestion:

you wrote:

The extracts displayed moderate and low capacity to inhibit AChE and BuChE, respectively. The extracts had reduced capacity to inhibit amylase and glucosidase, except for the ethanol fruits extracts, which coud significantly inhibit the latter enzyme. Except for the water extracts from fruits, samples had a strong inhibitory capacity towards tyrosinase. Quercetin and kaempferol glycosides fit well into the enzymatic pocket of the latter enzyme. ADME studies suggests that these molecules exhibit low
oral absorption, no activity in the CNS and, in general, unfavourably pharmacokinetic properties.

I prefer:

The extracts displayed moderate and low capacity to inhibit AChE and BuChE, respectively. The extracts had reduced capacity to inhibit amylase and glucosidase, except for the ethanol fruits extracts, which could significantly inhibit the latter enzyme. ADME studies suggests that these molecules also exhibit low oral absorption. On the other hand, interestingly, except for the water extracts from fruits, tested samples had a strong inhibitory capacity towards tyrosinase containing quercetin and kaempferol glycosides that fit well into the enzymatic pocket of the latter enzyme. Samples had reduced....

Author Response

Reviewer #3

Comment #1. Some suggestions:

194: extracts

234: latter (the second mentioned before) or later (metabolites)?

256: ... organs of the studied species

284-285: ... may play a role in the anti-tyrosinase activity of the extracts

285: check special characters

300: specifically

301: delete: This suggests that the extracts may not serve as targets for further studies aimimg its use in therapeutics targeting such health disorders.

306: before In fact insert: . and correct applications

458: delete D. 

Answer. We are grateful to the reviewer for these valuable corrections. All the corrections were made in the revised manuscript.

Comment #2. From 499: just a suggestion:

you wrote:

The extracts displayed moderate and low capacity to inhibit AChE and BuChE, respectively. The extracts had reduced capacity to inhibit amylase and glucosidase, except for the ethanol fruits extracts, which coud significantly inhibit the latter enzyme. Except for the water extracts from fruits, samples had a strong inhibitory capacity towards tyrosinase. Quercetin and kaempferol glycosides fit well into the enzymatic pocket of the latter enzyme. ADME studies suggests that these molecules exhibit loworal absorption, no activity in the CNS and, in general, unfavourably pharmacokinetic properties.

I prefer:

The extracts displayed moderate and low capacity to inhibit AChE and BuChE, respectively. The extracts had reduced capacity to inhibit amylase and glucosidase, except for the ethanol fruits extracts, which could significantly inhibit the latter enzyme. ADME studies suggests that these molecules also exhibit low oral absorption. On the other hand, interestingly, except for the water extracts from fruits, tested samples had a strong inhibitory capacity towards tyrosinase containing quercetin and kaempferol glycosides that fit well into the enzymatic pocket of the latter enzyme. Samples had reduced....

Answer. We totally agree with the reviewer suggestion, and the text was modified accordingly.